# CONVOLUTIONAL MESH AUTOENCODERS FOR 3D FACE REPRESENTATION

## ABSTRACT

Convolutional neural networks (CNNs) have achieved state of the art performance on recognizing and representing audio, images, videos and 3D volumes; that is, domains where the input can be characterized by a regular graph structure. However, generalizing CNNs to irregular domains like 3D meshes is challenging. Importantly, training data for 3D meshes is often limited making it difficult to train deep models. To address this, we generalize convolutional autoencoders to mesh surfaces using a model with relatively few parameters. We perform spectral decomposition of meshes and apply convolutions directly in frequency space. In addition, we use max pooling and introduce upsampling within the network to represent meshes in a low dimensional space. We construct a complex dataset of 20,466 high resolution meshes with extreme facial expressions and encode it using our Convolutional Mesh Autoencoder. Despite limited training data, our method outperforms state-of-the-art PCA models of faces with 50% lower error, while using 75% fewer parameters.

## 1  INTRODUCTION

Convolutional neural networks (LeCun, 1989) have achieved state of the art performance in a large number of problems in computer vision (Krizhevsky et al., 2012; He et al., 2016), natural language processing (Mikolov et al., 2013) and speech processing (Graves et al., 2013). In recent years, CNNs have also emerged as rich models for generating both images (Goodfellow et al., 2014; Oord et al., 2016) and audio (van den Oord et al., 2016). These successes may be attributed to the multi-scale hierarchical structure of CNNs that allows them to learn translational-invariant localized features. Since the learned filters are shared across the global domain, the number of filter parameters is independent of the domain size. We refer the reader to Goodfellow et al. (2016) for a comprehensive overview of deep learning methods and the recent developments in the field.

Despite the recent success, CNNs have mostly been successful in Euclidean domains with grid-based structured data. In particular, most applications of CNNs deal with regular data structures such as images, videos, text and audio, while the generalization of CNNs to irregular structures like graphs and meshes is not trivial. Extending CNNs to graph structures and meshes has only recently drawn significant attention (Bruna et al., 2013; Defferrard et al., 2016; Bronstein et al., 2017). Following the work of Defferrard et al. (2016) on generalizing the CNNs on graphs using fast Chebyshev filters, we introduce a *convolutional mesh autoencoder architecture* for realistically representing high-dimensional meshes of 3D human faces and heads.

The human face is highly variable in shape as it is affected by many factors such as age, gender, ethnicity etc. The face also deforms significantly with expressions. The existing state of the art 3D face representations mostly use linear transformations (Tewari et al., 2017; Li et al., 2017; Thies et al., 2015) or higher-order tensor generalizations (Vlasic et al., 2005; Brunton et al., 2014a). While these linear models achieve state of the art results in terms of realistic appearance and Euclidean reconstruction error, we show that CNNs can perform much better at capturing highly non-linear extreme facial expressions with many fewer model parameters.

One challenge of training CNNs on 3D facial data is the limited size of current datasets. Here we demonstrate that, since these networks have fewer parameters than traditional linear models, they can be effectively learned with limited data. This reduction in parameters is attributed to the locally invariant convolutional filters that can be shared on the surface of the mesh. Recent work has

exploited thousands of 3D scans and 4D scan sequences for learning detailed models of 3D faces (Cosker et al., 2011; Yin et al., 2006; 2008; Savran et al., 2008; Cao et al., 2014). The availability of this data enables us to a learn rich non-linear representation of 3D face meshes that can not be captured easily by existing linear models.

In summary, our work introduces a convolutional mesh autoencoder suitable for 3D mesh processing. Our main contributions are:

- We introduce a mesh convolutional autoencoder consisting of mesh downsampling and mesh upsampling layers with fast localized convolutional filters defined on the mesh surface.
- We use the mesh autoencoder to accurately represent 3D faces in a low-dimensional latent space performing 50% better than a PCA model that is used in state of the art methods (Tewari et al., 2017) for face representation.
- Our autoencoder uses up to 75% fewer parameters than linear PCA models, while being more accurate on the reconstruction error.
- We provide 20,466 frames of highly detailed and complex 3D meshes from 12 different subjects for a range of extreme facial expressions along with our code for research purposes. Our data and code is located at `http://withheld.for.review`.

This work takes a step towards the application of CNNs to problems in graphics involving 3D meshes. Key aspects of such problems are the limited availability of training data and the need for realism. Our work addresses these issues and provides a new tool for 3D mesh modeling.

## 2 RELATED WORK

**Mesh Convolutional Networks.** Bronstein et al. (2017) give a comprehensive overview of generalizations of CNNs on non-Euclidean domains, including meshes and graphs. Masci et al. (2015) defined the first mesh convolutions by locally parameterizing the surface around each point using geodesic polar coordinates, and defining convolutions on the resulting angular bins. In a follow-up work, Boscaini et al. (2016) parametrized local intrinsic patches around each point using anisotropic heat kernels. Monti et al. (2017) introduced $d$-dimensional pseudo-coordinates that defined a local system around each point with weight functions. This method resembled the intrinsic mesh convolution of Masci et al. (2015) and Boscaini et al. (2016) for specific choices of the weight functions. In contrast, Monti et al. used Gaussian kernels with trainable mean vector and covariance matrix as weight functions.

In other work, Verma et al. (2017) presented dynamic filtering on graphs where filter weights depend on the input data. The work however did not focus on reducing the dimensionality of graphs or meshes. Yi et al. (2017) also presented a spectral CNN for labeling nodes which did not involve any dimensionality reduction of the meshes. Sinha et al. (2016) and Maron et al. (2017) embedded mesh surfaces into planar images to apply conventional CNNs. Sinha et al. used a robust spherical parametrization to project the surface onto an octahedron, which is then cut and unfolded to form a squared image. Maron et al. (2017) introduced a conformal mapping from the mesh surface into a flat torus.

Although, the above methods presented generalizations of convolutions on meshes, they do not use a structure to reduce the meshes to a low dimensional space. The proposed mesh autoencoder efficiently handles these problems by combining the mesh convolutions with efficient mesh-downsampling and mesh-upsampling operators.

**Graph Convolutional Networks.** Bruna et al. (2013) proposed the first generalization of CNNs on graphs by exploiting the connection of the graph Laplacian and the Fourier basis (see Section 3 for more details). This lead to spectral filters that generalize graph convolutions. Boscaini et al. (2015) extended this using a windowed Fourier transform to localize in frequency space. Henaff et al. (2015) built upon the work of Bruna et al. by adding a procedure to estimate the structure of the graph. To reduce the computational complexity of the spectral graph convolutions, Defferrard et al. (2016) approximated the spectral filters by truncated Chebyshev poynomials which avoids explicitly computing the Laplacian eigenvectors, and introduced an efficient pooling operator for graphs. Kipf & Welling (2016) simplified this using only first-order Chebyshev polynomials.

However, these graph CNNs are not directly applied to 3D meshes. Our mesh autoencoder is most similar to Defferrard et al. (2016) with truncated Chebyshev polynomials along with the efficient graph pooling. In addition, we define mesh upsampling layer to obtain a complete mesh autoencoder structure and use our model for representation of highly complex 3D faces obtained state of the art results in realistic modeling of 3D faces.

**Learning Face Representations.** Blanz & Vetter (1999) introduced the first generic representation for 3D faces based on principal component analysis (PCA) to describe facial shape and texture variations. We also refer the reader to Brunton et al. (2014b) for a comprehensive overview of 3D face representations.

Representing facial expressions with linear spaces has given state-of-the-art results till date. The linear expression basis vectors are either computed using PCA (e.g. Amberg et al., 2008; Breidt et al., 2011; Li et al., 2017; Tewari et al., 2017; Yang et al., 2011), or are manually defined using linear blendshapes (e.g. Thies et al., 2015; Li et al., 2010; Bouaziz et al., 2013). Multilinear models (Vlasic et al., 2005), i.e. higher-order generalizations of PCA are also used to model facial identity and expression variations. In such methods, the model parameters globally influence the shape, i.e. each parameter affects all the vertices of the face mesh. To capture localized facial details, Neumann et al. (2013) and Ferrari et al. (2015) used sparse linear models. Brunton et al. (2014a) used a hierarchical multiscale approach by computing localized multilinear models on wavelet coefficients.

Brunton et al. (2014a) also used a hierarchical multi-scale representation, but their method does not use shared parameters across the entire domain. Jackson et al. (2017) use a volumetric face representation in their CNN-based framework. In contrast to existing face representation methods, our mesh autoencoder uses convolutional layers to represent faces with significantly fewer parameters. Since, it is defined completely on the mesh space, we do not have memory constraints which effect volumetric convolutional methods for representing 3D models.

## 3 MESH OPERATORS

We define a face mesh as a set of vertices and edges $\mathcal{F} = (\mathcal{V}, A)$, with $|\mathcal{V}| = n$ vertices that lie in 3D Euclidean space, $\mathcal{V} \in \mathbb{R}^{n \times 3}$. The sparse adjacency matrix $A \in \{0,1\}^{n \times n}$ represents the edge connections, where $A_{ij} = 1$ denotes an edge connecting vertices $i$ and $j$, and $A_{ij} = 0$ otherwise. The non-normalized graph Laplacian is defined as $L = D - A$ (Chung, 1997), with the diagonal matrix $D$ that represents the degree of each vertex in $\mathcal{V}$ as $D_{ii} = \sum_j A_{ij}$.

The Laplacian can be diagonalized by the Fourier basis $U \in \mathbb{R}^{n \times n}$ (as $L$ is a real symmetric matrix) as $L = U \Lambda U^T$, where the columns of $U = [u_0, u_1, ..., u_{n-1}]$ are the orthogonal eigenvectors of $L$, and $\Lambda = diag([\lambda_0, \lambda_1, ..., \lambda_{n-1}]) \in \mathbb{R}^{n \times n}$ is a diagonal matrix with the associated real, non-negative eigenvalues. The graph Fourier transform (Chung, 1997) of the mesh vertices $x \in \mathbb{R}^{n \times 3}$ is then defined as $x_\omega = U^T x$, and the inverse Fourier transform as $x = U x_\omega$, respectively.

**Fast spectral convolutions.** The convolution operator $*$ can be defined in Fourier space as a Hadamard product, $x * y = U((U^T x) \odot (U^T y))$. This is computationally expensive with large number of vertices. The problem is addressed by formulating mesh filtering with a kernel $g_\theta$ using a recursive Chebyshev polynomial (Hammond et al., 2011; Defferrard et al., 2016). The filter $g_\theta$ is parametrized as a Chebyshev polynomial of order $K$ given by

$$g_\theta(L) = \sum_{k=0}^{K-1} \theta_k T_k(\tilde{L}),$$ (1)

where $\tilde{L} = 2L/\lambda_{max} - I_n$ is the scaled Laplacian, the parameter $\theta \in \mathbb{R}^K$ is a vector of Chebyshev coefficients, and $T_k \in \mathbb{R}^{n \times n}$ is the Chebyshev polynomial of order $k$ that can be computed recursively as $T_k(x) = 2x T_{k-1}(x) - T_{k-2}(x)$ with $T_0 = 1$ and $T_1 = x$. The spectral convolution can then be defined as (Defferrard et al., 2016)

$$y_j = \sum_{i=1}^{F_{in}} g_{\theta_{i,j}}(L) x_i \in \mathbb{R}^n$$ (2)

| Layer | Input Size | Output Size |
|---|---|---|
| Convolution | $8192 \times 3$ | $8192 \times 16$ |
| Downsampling | $8192 \times 16$ | $2048 \times 16$ |
| Convolution | $2048 \times 16$ | $2048 \times 16$ |
| Downsampling | $2048 \times 16$ | $512 \times 16$ |
| Convolution | $512 \times 16$ | $512 \times 16$ |
| Downsampling | $512 \times 16$ | $128 \times 16$ |
| Convolution | $128 \times 16$ | $128 \times 32$ |
| Downsampling | $128 \times 32$ | $32 \times 32$ |
| Fully Connected | $32 \times 32$ | $8$ |

Table 1: Encoder architecture

| Layer | Input Size | Output Size |
|---|---|---|
| Fully Connected | $8$ | $32 \times 32$ |
| Upsampling | $32 \times 32$ | $128 \times 32$ |
| Convolution | $128 \times 32$ | $128 \times 32$ |
| Upsampling | $128 \times 32$ | $512 \times 32$ |
| Convolution | $512 \times 32$ | $512 \times 16$ |
| Upsampling | $512 \times 16$ | $2048 \times 16$ |
| Convolution | $2048 \times 16$ | $2048 \times 16$ |
| Max Pool | $2048 \times 16$ | $8192 \times 16$ |
| Convolution | $8192 \times 16$ | $8192 \times 3$ |

Table 2: Decoder architecture

that computes the $j^{th}$ feature of $y \in \mathbb{R}^{n \times F_{out}}$. The input $x \in \mathbb{R}^{n \times F_{in}}$ has $F_{in}$ features. The input face mesh has $F_{in} = 3$ features corresponding to its 3D vertex positions. Each convolutional layer has $F_{in} \times F_{out}$ vectors of Chebyshev coefficients $\theta_{i,j} \in \mathbb{R}^K$ as trainable parameters.

**Mesh Sampling** The mesh sampling operators define the downscaling and upscaling of the mesh features in a neural net. We perform the in-network downsampling of a mesh with $m$ vertices using transform matrices $Q_d \in \{0, 1\}^{n \times m}$, and upsampling using $Q_u \in \mathbb{R}^{m \times n}$ where $m > n$.

The downsampling is obtained by contracting vertex pairs iteratively that maintains surface error approximations using quadric matrices (Garland & Heckbert, 1997). The vertices after downsampling are a subset of the original mesh vertices $\mathcal{V}_d \subset \mathcal{V}$. Each weight $Q_d(p, q) \in \{0, 1\}$ denotes if $q$-th vertex is kept during downsampling $Q_d(p, q) = 1$, or discarded where $Q_d(p, q) = 0 \; \forall p$. Since a loss-less downsampling and upsampling is not feasable for general surfaces, the upsampling matrix is built during downsampling. Vertices kept during downsampling are kept during upsampling $Q_u(q, p) = 1$ iff $Q_d(p, q) = 1$.

Vertices $v_q \in \mathcal{V}$ discarded during downsampling where $Q_d(p, q) = 0 \; \forall p$, are mapped into the downsampled mesh surface. This is done by projecting $v_q$ into the closest triangle $(i, j, k)$ in the downsampled mesh, denoted by $\widetilde{v}_p$, and computing the Barycentric coordinates as $\widetilde{v}_p = w_i v_i + w_j v_j + w_k v_k \; (v_i, v_j, v_k \in \mathcal{V}_d)$. The weights are then updated in $Q_u$ as $Q_u(q, i) = w_i$, $Q_u(q, j) = w_j$, and $Q_u(q, k) = w_k$, and $Q_u(q, l) = 0$ otherwise.

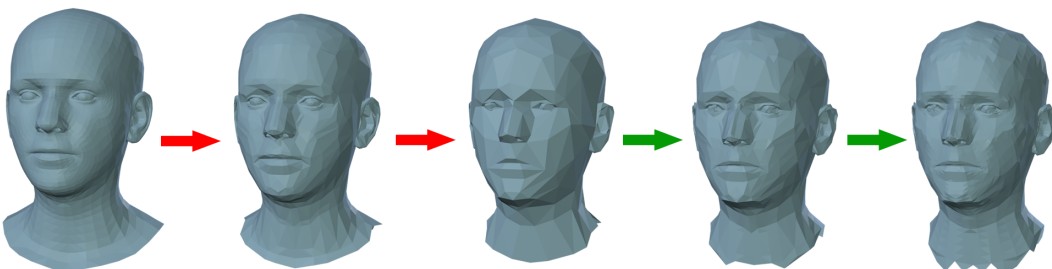

Figure 1: The effect of downsampling (red arrows) and upsampling (green arrows) on 3D face meshes. The reconstructed face after upsampling maintains the overall structure but most of the finer details are lost.

# 4 MESH AUTOENCODER

Now that we have defined the basic operations needed for our neural network in Section 3, we can construct the architecture of the convolutional mesh autoencoder. The structure of the encoder is shown in Table 1. The encoder consists of 4 Chebyshev convolutional filters with $K = 6$ Chebyshev polynomials. Each of the convolutions is followed by a biased ReLU (Glorot et al., 2011). The downsampling layers are interleaved between convolution layers. Each of the downsampling layers reduce the number of mesh vertices by 4 times. The encoder transforms the face mesh from $\mathbb{R}^{n \times 3}$ to an 8 dimensional latent vector using a fully connected layer at the end.

The structure of the decoder is shown in Table 2. The decoder similarly consists of a fully connected layer that transforms the latent vector from $\mathbb{R}^8$ to $\mathbb{R}^{32\times32}$ that can be further upsampled to reconstruct the mesh. Following the decoder's fully connected layer, 4 convolutional layers with interleaved upsampling layers generated a 3D mesh in $\mathbb{R}^{8192\times3}$. Each of the convolutions is followed by a biased ReLU similar to the encoder network. Each upsampling layer increases the numbers of vertices by 4x. The Figure 2 shows the complete structure of our mesh autoencoder.

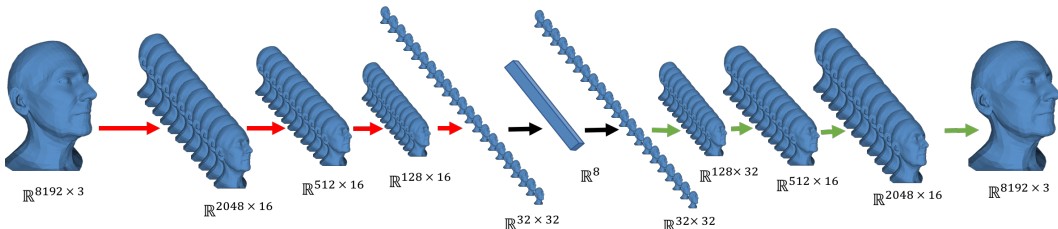

Figure 2: The architecture of the Convolutional Mesh Autoencoder. The red and green arrows indicate downsampling and upsampling layers respectively. The output space of each of the layers in denoted under it. Faces in intermediate layers are only for visualization.

**Training.** We train our autoencoder for 300 epochs with a learning rate of 8e-3 with a learning rate decay of 0.99 every epoch. We use stochastic gradient descent with a momentum of 0.9 to optimize the L1 loss between predicted mesh vertices and the ground truth samples. We use a regularization on the weights of the network using weight decay of 5e-4. The convolutions use Chebyshev filtering with $K = 6$.

## 5 EXPERIMENTS

**Facial Expression Dataset.** Our dataset consists of 12 classes of extreme expressions from 12 different subjects. These expressions are highly complex and uncorrelated with each other. The expressions in our dataset are – bareteeth, cheeks in, eyebrow, high smile, lips back, lips up, mouth down, mouth extreme, mouth middle, mouth side and mouth up. The number of frames of each sequence is shown in Table 3.

The data is captured at 60fps with a multi-camera active stereo system (3dMD LLC, Atlanta) with six stereo camera pairs, five speckle projectors, and six color cameras. Our dataset contains 20,466 3D Meshes, each with about 120,000 vertices. The data is pre-processed using a sequential mesh registration method (Li et al., 2017) to reduce the data dimensionality to 5023 vertices. We pre-process the data by adding fake vertices to increase the number of vertices to 8192. This enables pooling and upsampling of the mesh across the layers with a constant factor.

**Implementation details** We use Tensorflow (Abadi et al., 2016) for our network implementation. We use Scikit-learn (Pedregosa et al., 2011) for computing PCA coefficients. Training each network takes about 8 hours on a single Nvidia Tesla P100 GPU. Each of the models is trained for 300 epochs with a batch size of 16.

| **Sequence** | bareteeth | cheeks in | eyebrow | high smile | lips back | lips up |
|---|---|---|---|---|---|---|
| **# Frames** | 1946 | 1396 | 2283 | 1878 | 1694 | 1511 |

| **Sequence** | mouth down | mouth extreme | mouth middle | mouth open | mouth side | mouth up |
|---|---|---|---|---|---|---|
| **# Frames** | 2363 | 793 | 1997 | 674 | 1778 | 2153 |

Table 3: Length of different expression sequences

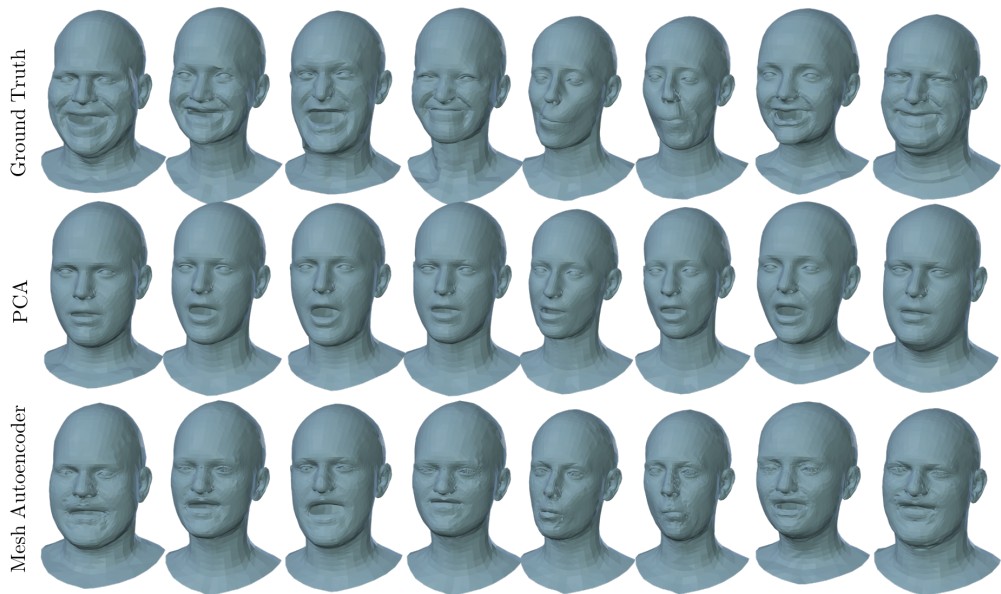

Figure 3: Qualitative results for mesh reconstruction on test set of interpolation experiment.

|  | Mean Error | Median Error | # Parameters |
|---|---|---|---|
| PCA | $2.694 \pm 2.330$ | 2.002 | 120,552 |
| Mesh Autoencoder | **1.578 ± 1.546** | **1.110** | **33,856** |

Table 4: Performance comparison with different error metrics on interpolation experiment. Mean error is of the form $[\mu \pm \sigma]$ with mean Euclidean distance $\mu$ and standard deviation $\sigma$. The median error and number of parameters in each model are also shown. All errors are in millimeters (mm).

## 5.1 INTERPOLATION EXPERIMENT

We evaluate the performance of our model based on its ability to interpolate the training data and extrapolate outside its space. We compare the performance of our model with a PCA model. We consistently use an 8-dimensional latent space to encode the face mesh using both the PCA model and the Mesh Autoencoder. Thus, the encoded latent vectors lie in $\mathbb{R}^8$. Meanwhile, the number of parameters in our model is much smaller than PCA model (Table 4).

In order to evaluate the interpolation capability of the autoencoder, we split the dataset in training and test samples in the ratio of 1:9. The test samples are obtained by picking consecutive frames of length 10 uniformly at random across the sequences. We train our autoencoder for 300 epochs and evaluate it on the test set. We use mean Euclidean distance for comparison with the PCA method. The mean Euclidean distance of $N$ test mesh samples with $n$ vertices each is given by

$$\mu = \frac{1}{nN} \sum_{i=1}^{N} \sum_{j=1}^{n} ||x_{ij} - \hat{x}_{ij}||_2 \tag{3}$$

where $x_{ij}, \hat{x}_{ij} \in \mathbb{R}^3$ are $j$-th vertex predictions and ground truths respectively corresponding to $i$-th sample. Table 4 shows the mean Euclidean distance along with standard deviation in the form $[\mu \pm \sigma]$. The median error is also shown in the table. We show a performance improvement, as high as 50% over PCA models for capturing these highly non linear facial expressions. At the same time, the number of parameters in the CNN is about 75% fewer than the PCA model as shown in Table 4. Visual inspection of our qualitative results in Figure 3 show that our reconstructions are more realistic and are effective in capturing extreme facial expressions. We also show the histogram of cumulative errors in Figure 4a. We observe that Mesh Autoencoder has about 76.9% of the vertices within an Euclidean error of 2 mm, as compared to 51.7% for the PCA model.

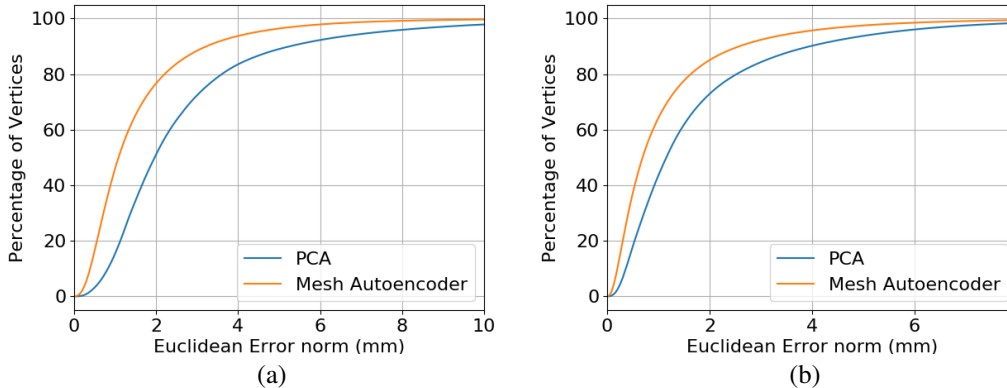

Figure 4: Cumulative Euclidean error between PCA model and Mesh Autoencoder for Interpolation(a) and Extrapolation(b) experiments.

## 5.2 Extrapolation Experiment

To measure generalization of our model, we compare the performance of our model with a PCA model and FLAME (Li et al., 2017). For comparison, we train the expression and jaw model of FLAME with our dataset. The FLAME reconstructions are obtained with with latent vector size of 16 with 8 components each for encoding identity and expression. The latent vectors encoded using PCA model and Mesh autoencoder have a size of 8.

We evaluate the generalization capability of the Mesh Autoencoder by attempting to reconstruct the expressions that are completely unseen by our model. We split our dataset by completely excluding one expression set from all the subjects of the dataset. We test our Mesh Autoencoder on the excluded expression as the test set. We compare the performance of our model with PCA and FLAME using the same mean Euclidean distance. We perform 12 cross validation experiments, one for each expression as shown in Table 5. For each experiment, we run our training procedure ten times initializing the weights at random. We pick the best performing network for comparison.

We compare the results using mean Euclidean distance and median error metric in Table 5. Our method performs better than PCA model and FLAME (Li et al., 2017) on all expression sequences. We show the qualitative results in Figure 5. Our model performs much better on these extreme expressions. We show the cumulative euclidean error histogram in Figure 4b. For a 2 mm accuracy, Mesh Autoencoder captures 84.9% of the vertices while the PCA model captures 73.6% of it.

## 5.3 Ablation Experiments

The FLAME model Li et al. (2017) uses several PCA-models to represent expression, jaw motion, face identity etc. We evaluate the performance of mesh autoencoders by replacing the expression model of FLAME by our autoencoder. We compare the reconstruction errors with the original FLAME model. We run our experiment by varying the size of the latent vector for encoding. We show the comparisons in Table 6.

## 5.4 Discussion

While our convolutional Mesh Autoencoder leads to a representation that generalizes better for unseen 3D faces than PCA with much fewer parameters, our model has several limitations. Our network is restricted to learning face representation for a fixed topology, i.e., all our data samples needs to have the same adjacency matrix, $A$. The mesh sampling layers are also based on this fixed adjacency matrix $A$, which defines only the edge connections. The adjacency matrix does not take in to account the vertex positions thus affecting the performance of our sampling operations. In future, we would like to incorporate this information into our learning framework.

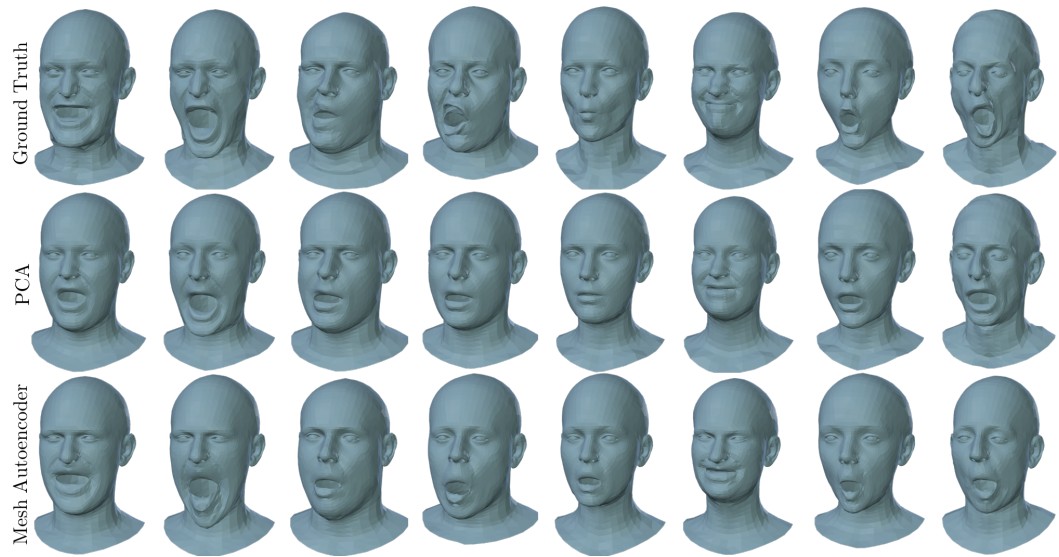

Figure 5: Qualitative results on test set of extrapolation experiments. The expression of the test set was completely excluded from all subjects.

| Sequence | Mesh Autoencoder Mean Error | Median | PCA Mean Error | Median | FLAME (Li et al., 2017) Mean Error | Median |
|---|---|---|---|---|---|---|
| bareteeth | **1.376±1.536** | **0.856** | 1.957±1.888 | 1.335 | 2.002±1.456 | 1.606 |
| cheeks in | **1.288±1.501** | **0.794** | 1.854±1.906 | 1.179 | 2.011±1.468 | 1.609 |
| eyebrow | **1.053±1.088** | **0.706** | 1.609±1.535 | 1.090 | 1.862±1.342 | 1.516 |
| high smile | **1.205±1.252** | **0.772** | 1.841±1.831 | 1.246 | 1.960±1.370 | 1.625 |
| lips back | **1.193±1.476** | **0.708** | 1.842±1.947 | 1.198 | 2.047±1.485 | 1.639 |
| lips up | **1.081±1.192** | **0.656** | 1.788±1.764 | 1.216 | 1.983±1.427 | 1.616 |
| mouth down | **1.050±1.183** | **0.654** | 1.618±1.594 | 1.105 | 2.029±1.454 | 1.651 |
| mouth extreme | **1.336±1.820** | **0.738** | 2.011±2.405 | 1.224 | 2.028±1.464 | 1.613 |
| mouth middle | **1.017±1.192** | **0.610** | 1.697±1.715 | 1.133 | 2.043±1.496 | 1.620 |
| mouth open | **0.961±1.127** | **0.583** | 1.612±1.728 | 1.060 | 1.894±1.422 | 1.544 |
| mouth side | **1.264±1.611** | **0.730** | 1.894±2.274 | 1.132 | 2.090±1.510 | 1.659 |
| mouth up | **1.097±1.212** | **0.683** | 1.710±1.680 | 1.159 | 2.067±1.485 | 1.680 |

Table 5: Quantitative evaluation of Extrapolation experiment. The training set consists of the rest of the expressions. Mean error is of the form $[\mu \pm \sigma]$ with mean Euclidean distance $\mu$ and standard deviation $\sigma$. The median error and number of frames in each expression sequnece is also shown. All errors are in millimeters (mm).

The amount of data for high resolution faces is very limited. We believe that generating more of such data with high variability between faces would improve the performance of Mesh Autoencoders for 3D face representations. The data scarcity also limits our ability to learn models that can be trained for superior performance at higher dimensional latent space. The data scarcity also produces noise in some reconstructions.

# 6 CONCLUSION

We have introduced a generalization of convolutional autoencoders to mesh surfaces with mesh downsampling and upsampling layers combined with fast localized convolutional filters in spectral space. The locally invariant filters that are shared across the surface of the mesh significantly reduce the number of filter parameters in the network. While the autoencoder is applicable to any class of mesh objects, we evaluated its quality on a dataset of realistic extreme facial expressions. The local

| #dim of $z$ | FLAME++ | | FLAME Li et al. (2017) | |
|---|---|---|---|---|
| | Mean Error | Median | Mean Error | Median |
| 2 | **0.610±0.851** | **0.317** | 0.668±0.876 | 0.371 |
| 4 | **0.509±0.746** | **0.235** | 0.589±0.803 | 0.305 |
| 6 | **0.464±0.711** | **0.196** | 0.525±0.743 | 0.252 |
| 8 | **0.432±0.681** | **0.169** | 0.477±0.691 | 0.217 |
| 10 | **0.421±0.664** | **0.162** | 0.439±0.655 | 0.193 |
| 12 | **0.388±0.630** | **0.139** | 0.403±0.604 | 0.172 |
| 14 | 0.371±0.605 | **0.128** | **0.371±0.567** | 0.152 |
| 16 | 0.372±0.611 | **0.125** | **0.351±0.543** | 0.139 |

Table 6: Comparison of FLAME and FLAME++. FLAME++ is obtained by replacing expression model of FLAME with our mesh autoencoder. All errors are in millimeters (mm).

convolutional filters capture a lot of surface details that are generally missed in linear models like PCA while using 75% fewer parameters. Our Mesh Autoencoder outperforms the linear PCA model by 50% on interpolation experiments and generalizes better on completely unseen facial expressions.

Face models are used in a large number of applications in computer animations, visual avatars and interactions. In recent years, a lot of focus has been given to capturing highly detailed static and dynamic facial expressions. This work introduces a direction in modeling these high dimensional face meshes that can be useful in a range of computer graphics applications.

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
