# OpenReview forum: "Convolutional Mesh Autoencoders for 3D Face Representation"
_ICLR.cc/2018/Conference — Reject_

### Official Review · AnonReviewer1 · 2017-11-22
**no human evaluation?**

**Rating:** 6
**Confidence:** 3

**Review:**

The paper is generally clear, and proposes to use a convolutional autoencoder based on 3D meshes. The novelty here how the problem is formulated.

Pros:
- Interesting formulation. I have not seen this particular setup for processing meshes with neural networks in an autoencoder setting.
- This work collected a new dataset for 3D face expression representation, which is great (the state of 3D face databases which are available to researchers is very limited, so this is a step in the right direction).

Cons:
- The visual depiction of the auto-encoded meshes looks a bit strange. In particular, they exhibit some high frequency artefacts. These do not appear in the smoother PCA version. From a human standpoint, in those cases, the smoother meshes would in fact be preferable. I did not see a discussion about this, given that such problems are not captured by the metrics.
- I am a bit confused by the requirement that all meshes need to have the same adjacency matrix. Does this mean that you need to convert the raw meshes coming from the 3D camera into a particular topology before you can use this algorithm? If yes, this seems like a rather large limitation.
- Regarding the evaluation, you wrote:"In order to evaluate the interpolation capability of the autoencoder, we split the dataset in training and test samples in the ratio of 1:9. The test samples are obtained by picking consecutive frames of
length 10 uniformly at random across the sequences. " - To me this is very unclear. You have very few sequeces/subjects. Did  you split by *subject*? I think this is CRUCIAL, and a lot of the results hinge on this answer.

General Questions

I am  wondering how come you didn't consider a geometry image representation of the meshes, and went for a slightly more general, and yet very confined alternative (the adjacency requirement, which in some sense is the same type of constraint as geometry images). On geometry images, in particular it would be possible to apply standard convolutional architectures without any special processing.

Another question that I had is why use a L1 loss when in the evaluation you're using L2? It would make a lot of sense to use the same loss as the evaluation metric (not to mention the properties of PCA).

---

> ### Author Response · Authors · 2018-01-05
> **Re:**
>
> Dataset splits: We  split the dataset by expressions. We captured expressions keeping in mind that each of them were largely uncorrelated. For cross-validation, we leave out one expression from all subjects that forms the test set.
>
> Topology: We register all the meshes to a regular topology and then learn the representation. This is widely used in shape modeling problems [3,4].
>
> Geometry Image representations: The geometry image representation uses unfolding of meshes on a plane. This works only for a face but not for the whole head containing face, eyes etc.
>
> Human Evaluation: We will run human evaluations and report it in the final version.
>
> Losses: We use L1 for training since it is more robust to outliers in the training dataset. We use L2 for testing, since it is consistent with other papers [3].
>
> [3] T. Li, T. Bolkart, M. J. Black, H. Li, and J. Romero. Learning a model of facial shape and expression from 4D scans. ACM Transactions on Graphics, 36(6), 2017.
> [4]Davies, Rhodri, Carole Twining, and Chris Taylor. Statistical models of shape: Optimisation and evaluation. Springer Science & Business Media, 2008.

---

### Official Review · AnonReviewer2 · 2017-11-25
**Authors extend [1] to form an auto-encoder CNN network for face mesh representation.**

**Rating:** 2
**Confidence:** 5

**Review:**

Paper summary:
Authors extend [1] to form an auto-encoder CNN network for face mesh representation. Face mesh graph is represented by Fourier basis of graph Laplacian and therefore convolution operator is defined in Fourier space. Chebyshev polynomial is used for faster computations. Max pooling on graph is done by using Graclus multilevel clustering algorithm. Binary tree generated in pooling layers are kept for unpooling layers in decoder network. Authors captured a new facial dataset for their evaluation and reported better results than PCA.

Positive points:
Authors tackle irregular data feature extraction and learning using CNNs which is a hot topic in deep learning.

Negative points:
Although proposed idea is interesting, paper has a number of critical problems. Firstly, experiments are the main weakness of the paper. Set of experiments does not prove claims of the paper.
- It is not clear how authors uses PCA to reconstruct faces in the test set.
- Authors do not compare to any state of the art on 3D face representation and reconstruction (e.g. [2]) using public datasets (e.g. BU-3DFE).
- How network behaves by introducing noise on vertices?
- What is the effect of network hyper-parameters?

Secondly, paper has a lack of novelty. It is a simple extension of [1] without considering and solving problems in [1]. Also, it is not mentioned what is the loss function to train the network. I suppose it is L2 norm loss, but it must be clear in the paper.

[1] M. Defferrard, X. Bresson, and P. Vandergheynst.  Convolutional neural networks on graphs with fast localized spectral filtering. In Advances in Neural Information Processing Systems, pp. 3844–3852, 2016.
[2] A. Brunton, T. Bolkart, and S. Wuhrer.  Multilinear wavelets: A statistical shape space for human faces. In European Conference on Computer Vision, pp. 297–312, 2014a.

After rebuttal:
The current version of the paper still needs significant amount of work regarding the experimental part.

---

> ### Author Response · Authors · 2018-01-05
> **Re:**
>
> Choice of Network Hyperparameters: We ran several trials to obtain the range of hyperparameters which perform better than others. The best way would be to run an exhaustive hyperparameter search, however, this is computationally not feasible since it would require training 100s of networks.
>
> L1 Loss: We use L1 loss and have clarified it in the revised version.
>
> Experiments: We added two experiments (Table 5,6) comparing to state of art model [3] and show better performance. We will run a denoising experiment and add it in the final version.
>
> PCA: We treat PCA like a simple linear autoencoder. We project a face mesh into the latent PCA space, and then reconstruct it using the PCA basis. Shape modeling methods [3] use such PCA formulations.
>
> [3] T. Li, T. Bolkart, M. J. Black, H. Li, and J. Romero. Learning a model of facial shape and expression from 4D scans. ACM Transactions on Graphics, 36(6), 2017.

---

### Official Review · AnonReviewer3 · 2017-11-27
**Not enough novelty / lack of experiments**

**Rating:** 4
**Confidence:** 4

**Review:**

This paper introduces a convolutional autoencoder for irregular graphs, specifically surfaces in the form of discrete meshes in 3D. The underlying technique that is used to operate on the irregular graph is spectral decomposition, which enables convolutions in the spectral domain.

The spectral convolution methods have been applied to mesh data structures for about 5 years now, as stated in the paper as well [Bruna et al. 2013], [Defferrard et al. 2016], [Bronstein et al. 2017], [Li et al. 2017] ... The paper states that it builds upon the formulation in [Defferrard et al. 2016] as explained in section 3.

Given the facts above, I am having a hard time to understand the novelty of this paper? Is it the "Mesh Upsampling" operation defined at the end of page 4? If that is the case, it is not demonstrated in the paper that it actually works. The main reason is that the original face mesh graph that goes into the convolution/downsampling
 operations is topologically preserved through the upconvolutions. This means that the upsampling operation is not really upsampling a "true" graph/mesh. The topology is already known, the upsampling just predicts a function on this topology.

 The choice of the face domain is also suspicious, since all faces are topologically the same graph (even though there are geometric variations). Convolutions/downsampling-convolutions/upsampling that are demonstrated in the paper basically boil down to function prediction on the same exact global graph. Face topologies are so regular that they can even be represented with a height map like geometry encoding in the image plane (See [1'] below).

 To demonstrate the "mesh/graph generation" capability truly, the authors need to experiment on novel topology generation. As is, the paper does not bear enough novelty on top of [Defferrard et al. 2016], or is not demonstrating it even if there exists any.

[1'] Z. Shu, E. Yumer, S. Hadap, K. Sunkavalli, E. Shechtman, D. Samaras. Neural Face Editing with Intrinsic Image Disentangling. CVPR 2017

MINOR:
First paragraph of Section 3:
- The definition of a mesh (F=(V,E,A)) is not correct: Both E and A essentially define the same connectivity. Base your definition on F=(V,E) or F=(V,A). Since you are using A later in the section, probably the latter makes more sense for you.

---

> ### Author Response · Authors · 2018-01-05
> **Re:**
>
> Choice of face domain and regular topology:
> Our work is motivated to be useful for the problem of representing 3D faces. Current practice in face modeling use linear or multilinear models for representing 3D faces given a regular topology. These models do not capture the non-linear nature of facial motion. Given the recent advances in deep learning, we use a non linear model to capture detailed facial deformations. Our model is also applicable for representing other classes of shapes such as bodies and hands. Using a predefined topology is a common practice in shape modeling [3,4].
>
> [3] T. Li, T. Bolkart, M. J. Black, H. Li, and J. Romero. Learning a model of facial shape and expression from 4D scans. ACM Transactions on Graphics, 36(6), 2017.
> [4]Davies, Rhodri, Carole Twining, and Chris Taylor. Statistical models of shape: Optimisation and evaluation. Springer Science & Business Media, 2008.

---

### Author Response · Authors · 2018-01-05
**Revisions, Experiments and Comparisons**

Dear Reviewers,
We thank you for your valuable comments. We have revised the paper, added further experiments and addressed your comments.

1. We introduced a novel sampling operator for upsampling and downsampling based on Qslim framework[1] in Section 3. This achieves better performance for sampling 3D meshes while preserving their structure (Figure 1 in the revised paper). We get better reconstructions and lower errors using this sampling technique. The earlier version of our paper used [2] for sampling which was a more general approach for graphs. This resulted in irregular surfaces with high frequency noise.

2. We added comparisons of our results with [3], a recently published state of art model for representing 3D faces (Table 5). We achieve better reconstruction errors than [3].

3. We added further experiments to evaluate the performance of [3] when parts of it are replaced by our trained mesh autoencoder (Table 6). We show that using replacing expression model of [3] with mesh autoencoder improves its performance.

4. Minor corrections are incorporated as suggested in the new version.

We address comments specific to the reviewers in the sections below.

[1] Michael Garland and Paul S Heckbert. Surface simplification using quadric error metrics. In Proceedings of the 24th annual conference on Computer graphics and interactive techniques, pp. 209–216. ACM Press/Addison-Wesley Publishing Co., 1997.
[2] M. Defferrard, X. Bresson, and P. Vandergheynst. Convolutional neural networks on graphs with fast localized spectral filtering. In Advances in Neural Information Processing Systems, pp. 3844–3852, 2016.
[3] T. Li, T. Bolkart, M. J. Black, H. Li, and J. Romero. Learning a model of facial shape and expression from 4D scans. ACM Transactions on Graphics, 36(6), 2017.
[4]Davies, Rhodri, Carole Twining, and Chris Taylor. Statistical models of shape: Optimisation and evaluation. Springer Science & Business Media, 2008.

---

### Decision · Program_Chairs · 2018-01-29
**ICLR 2018 Conference Acceptance Decision**

**Decision:**

Reject

**Comment:**

The paper was just not well enough received to warrant acceptance.